# Host-derived circular RNAs display proviral activities in Hepatitis C virus-infected cells

Tzu-Chun Chen[1,2⊚], Marc Tallo-Parra[3⊚], Qian M. Cao[1,2⊚], Sebastian Kadener[4], René Böttcher[3], Gemma Pérez-Vilaró[3], Pakpoom Boonchuen[5], Kunlaya Somboonwiwat[5], Juana Díez[3]*, Peter Sarnow[1,2]*

**1** Department of Microbiology & Immunology, Stanford University SOM, Stanford, California, United States of America, **2** Chan Zuckerberg Biohub, San Francisco, California, United States of America, **3** Molecular Virology Group, Department of Experimental and Health Sciences, Universitat Pompeu Fabra, Barcelona, Spain, **4** Department of Biology, Brandeis University, Waltham, Massachusetts, United States of America, **5** Department of Biochemistry, Chulalongkorn University, Bangkog, Thailand

⊚ These authors contributed equally to this work.
* juana.diez@upf.edu (JD); psarnow@stanford.edu (PS)

**Data Availability Statement:** Data are available from the GEO database with accession number GSE143300 (https://www.ncbi.nlm.nih.gov/geo/query/acc.cgi?acc=GSE143300).

## Abstract

Viruses subvert macromolecular pathways in infected host cells to aid in viral gene amplification or to counteract innate immune responses. Roles for host-encoded, noncoding RNAs, including microRNAs, have been found to provide pro- and anti-viral functions. Recently, circular RNAs (circRNAs), that are generated by a nuclear back-splicing mechanism of pre-mRNAs, have been implicated to have roles in DNA virus-infected cells. This study examines the circular RNA landscape in uninfected and hepatitis C virus (HCV)-infected liver cells. Results showed that the abundances of distinct classes of circRNAs were up-regulated or down-regulated in infected cells. Identified circRNAs displayed proviral effects. One particular up-regulated circRNA, circPSD3, displayed a very pronounced effect on viral RNA abundances in both hepatitis C virus- and Dengue virus-infected cells. Though circPSD3 has been shown to bind factor eIF4A3 that modulates the cellular nonsense-mediated decay (NMD) pathway, circPSD3 regulates RNA amplification in a pro-viral manner at a post-translational step, while eIF4A3 exhibits the anti-viral property of the NMD pathway. Findings from the global analyses of the circular RNA landscape argue that pro-, and likely, anti-viral functions are executed by circRNAs that modulate viral gene expression as well as host pathways. Because of their long half-lives, circRNAs likely play hitherto unknown, important roles in viral pathogenesis.

## Author summary

Usually, cells are infected by one or a few virus particles that carry genomes with limited expression capacity. Thus, the expression of viral genomes has to compete with a sea of cellular components that aid in viral translation, replication and virion production. Depending on their lifestyle, viruses have evolved to avoid or to subvert cellular pathways, especially those that display anti-viral functions. Host-derived circular RNA molecules have recently been discovered in the cytoplasm of cells, although, as-of-yet, few functions

**Funding:** This study was funded by the National Institutes of Health (https://www.nih.gov) (R01 AI06900011) and the Chan Zuckerberg BioHub to P.S, and NIH grants R01GM122406 and R01AG057700 to S.K. Funds for J.D. were received from the Spanish Ministry of Economy and Competitiveness (AEI/ MINECO/FEDER,UE) through grants BFU2016–80039-R and Unidad de Excelencia Maria de Maeztu funded by the MINECO (ref: MDM-2014–0370), and by a 2017 SGR 909 grant by the Secretaria d'Universitats i Recerca del Departament d'Economia i Coneixement of the Generalitat de Catalunya. The funders had no role in study design, data collection and analysis, decision to publish, or preparation of the manuscript.

**Competing interests:** The authors have declared that no competing interests exist.

have been assigned to them. Here, we describe alterations in the circular RNA landscape in hepatitis C virus-infected liver cells. Up-regulated and down-regulated circular RNAs were identified, and three of the upregulated RNAs were shown to promote HCV infection. One of them, circPSD3, inhibited viral RNA abundance at a post-translational step. Because circular RNAs are more stable than linear RNAs, they may have important functions during viral infection, dictating the outcomes of innate immune responses and viral pathogenesis.

## Introduction

Circular RNAs (circRNAs) were first detected in plant viral RNA pathogens, termed viroids [1]. Subsequently, it has shown that eukaryotic cells express circRNAs as well [2, 3]. Some circRNAs contain scrambled exons, which are produced by an abnormal splicing mechanism [4]. Recently, it has been shown that circRNA species are synthesized from pre-mRNAs by a nuclear back-splicing mechanism [5–8]. Most circRNAs are derived from exons that are flanked by inverted intronic sequences that can engage in base pair interactions, thereby enabling circularization of the RNA. Mediated by RNA binding proteins, the splice donor at the 3' end of the exon can juxtapose with the splice acceptor at its 5' end, allowing circularization of the RNA [9]. Salzman and colleagues [10] discovered a surprising feature of eukaryotic gene expression in linear RNA-depleted libraries. An abundance of thousands of circular RNAs were identified by their distinct 3'-to-5' junctions using algorithms such as findcirc or KNIFE [10–13]. These approaches revealed that approximately 5,000 to 25,000 circRNAs exist per cell, with 20% of transcribed genes producing unique circRNAs that can be expressed in a tissue-specific manner [10, 14].

However, little is known about their individual biological functions [6]. This is mainly due to the fact that their abundances range only between 1 and 10 copies per cell [15]. For example, a few circRNAs have been shown to stimulate RNA polymerase II transcription [16]. Curiously, two cytoplasmic circRNAs, CiRS-7 (in human) and Sry (in mouse), harbor multiple binding sites for microRNA-7 and microRNA-138, respectively, and likely regulate some aspect of miRNA function [17–19]. CircRNAs like circMbl [20] and circPABPN1 [21] sequester RNA binding proteins, while circANRIL [15] is involved in pre-rRNA processing. Ribosomal profiling and mass spectrometry analyses have shown that circRNAs Mbl [22] and ZNF609 [23] can be translated; however, functional roles for the translated proteins are not known. Global analyses of polysome-associated RNAs have suggested that most circRNAs are poorly translated in cultured cells [24], unless the circles contain internal ribosome entry sites [25] or methylated adenosine residues that can recruit the translation apparatus via circRNA-bound methyladenosine reader protein YTHDF3 [26]. Curiously, translation of circRNAs is much more prevalent in human organs [27]. Furthermore, extracellular association of circRNAs with 40-100nm small exosomes or 100-1000nm large microvesicles has suggested that circRNAs can spread from cell-to-cell [28]. Finally, their prolonged stability makes them useful as biomarkers for clinical disease [29, 30].

Because viruses usually subvert cellular pathways to modulate viral gene expression, several investigators have explored whether the cellular and viral circRNA landscapes are altered in cells infected by DNA viruses. Indeed, it has been found that cells infected by Epstein-Barr virus and Kaposi's sarcoma virus (KSHV) not only alter the cellular landscape of circRNAs, but also display circRNAs from viral pre-RNA transcripts [31–33]. Tagawa et al. noted that one host cell-derived circRNA, has_circ_0001400, inhibited the expression of the KSHV

latency gene LANA [32]. However, the roles for other circRNAs during DNA virus infections are not known. Much less is known about the roles for circRNAs during RNA virus infections. Li et al. [34] provided evidence that RNA virus infections can modulate circRNA abundances in infected cells. Specifically, these investigators noted that infection of cells with Vesicular stomatitis virus, a negative-stranded RNA virus, resulted in reduced abundances of host-derived circRNAs in the cytoplasm. It is thought that RNA binding protein NF90/NF110 is recruited to viral replication complexes in the cytoplasm, leaving fewer NF90/NF110 molecules in the nucleus to aid in the formation of circRNAs [34]. In the present study, we analyzed the global landscape of host-derived cellular circRNAs in hepatitis C virus (HCV)-infected cells. We discovered that circRNAs can display proviral activities. We focused our study one particular RNA, circPSD3, that modulates replication of different flaviviruses.

## Results

### Identification of circRNAs that display altered RNA abundances in HCV-infected cells

To examine the landscape of circRNA abundances during HCV infection, RNAs from mock- and JFH1 D183 (type 2)-infected cells were isolated from several distinct pools (Fig 1A). Libraries were constructed and subjected to RNA-sequencing (Fig 1A, see Materials and methods). CircRNAs, whose abundance was altered in infected relative to their linear counterparts, were identified by employing a custom pipeline for identification and quantification. Three publicly available programs for annotation of cirRNAs were used in combination with limma-voom (Fig 1A)[35], to determine whether changes in circRNA expression could be solely explained by comparable changes of the linear counterpart while accounting for within-sample variance.

Out of a total of 8176 circRNAs, 73 were detected as significantly altered in abundance relative to their linear counterparts (Fig 1B). To see if the host genes of these circRNAs shared common functions, we tested for enriched GO terms via the gprofiler2 R-package. However, this analysis yielded only one GO Cellular Component term ("nucleus"; p = 0.0395) for the genes showing decreased circRNA abundance, whereas genes with increased circRNA abundance were significantly enriched for the GO Molecular Function term "DNA replication origin binding" (p = 0.037).

Ten of which showed increased and 63 decreased abundances during HCV infection (Fig 1B). Several factors could alter the relative abundances of circRNAs during infection: changes in initiation or elongation of transcription, changes in back-splicing frequency, stability, or rates of cytoplasmic export. To focus our subsequent analyses, we chose circRNA candidates that displayed the biggest differences in abundance relative to their linear transcript. Two of the most up-regulated and one of the most down-regulated circRNAs were selected for further validation, and their predicted abundances were confirmed by qPCR in independent experiments (Fig 1C). The names and lengths of the examined circRNAs are listed in S1 Table. For functional studies, we focused on the roles for up-regulated circRNAs, because functional studies of down-regulated circRNAs involve the cumbersome task of over-expressing the circRNAs to their normal physiological concentrations. On the other hand, functional studies on up-regulated circRNAs can be performed more easily after depletion of the circRNAs by siRNAs that target the back-splicing junction sequences.

Data in Fig 2 show the effects of siRNA-mediated depletion of two up-regulated circRNAs, circEXOSC (Fig 2A, 2B and 2C) and circTIAL (Fig 2D, 2E and 2F), which were confirmed to be up-regulated during HCV infection (Fig 1C), on HCV infectivity. For both circEXOSC and circTIAL, three siRNAs were used that target overlapping sequences in the back-splicing

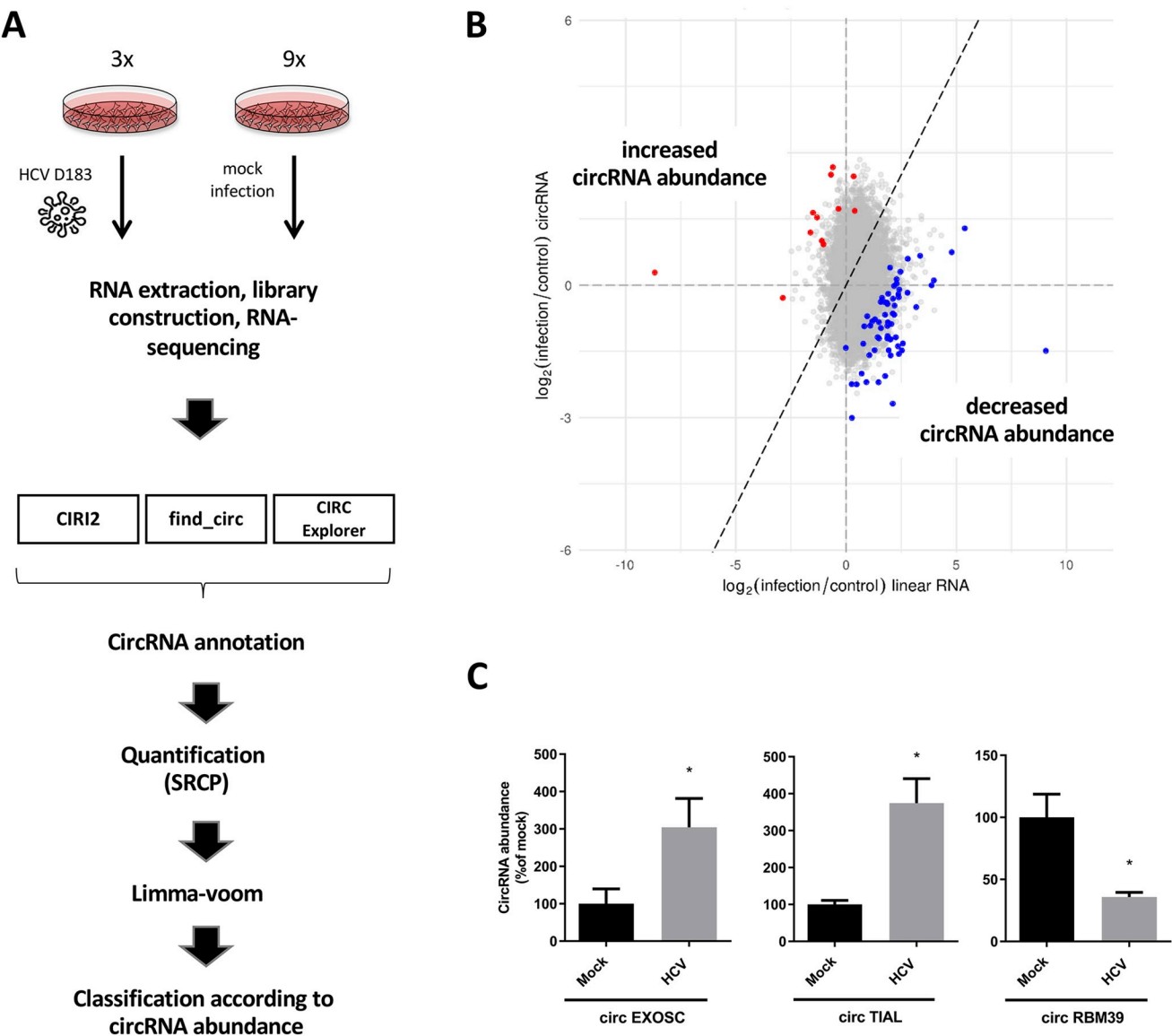

**Fig 1. HCV elicits differential expression of circRNAs.** (**A**) Experimental overview to identify HCV-induced circRNAs in infected and mock-infected cells. Sequencing data was analyzed with three different computational pipelines to identify circRNAs (CIRI2, find_circ, CIRC explorer) and quantified by SRCP. Statistical testing for differential circRNA expression was done using Limma-voom and the different circRNAs were classified according to their altered abundances. (**B**) Scatter plot of HCV-induced changes in circRNA abundances. Y- and X-axis represent the log2 fold change of circular and corresponding linear counterparts upon HCV infection, respectively. Dots represent individual candidate circRNAs with increased (red) or decreased (blue) abundances relative to their linear transcripts. Grey dots do not display changes in abundances. (**C**) qPCR-mediated validation of two up-regulated and one down-regulated top-candidate from the RNA-seq screening. Results were normalized to the housekeeping gene- Peptidylprolyl isomerase A (PPIA) RNA abundance and expressed as a percentage relative to the mock-infected sample. Bars represent mean values of three biological replicates (±SEM). Statistical significance was calculated using a T-test (* indicates $p < 0.05$).

junction sequence of the circRNAs but did not display sequence complementarity to other human sequences. The siRNAs directed against circEXOSC depleted the circRNAs with various efficiencies (Fig 2A), but did not affect the abundance of their linear counterparts (Fig 2B). Virus infectivity, measured by gaussia luciferase activity, was significantly diminished after depletion of circEXOSC in virus-infected cells (Fig 2C). SiRNA-mediated depletion of circ-TIAL was not as complete, with only one of the three siRNAs decreasing its abundance to a

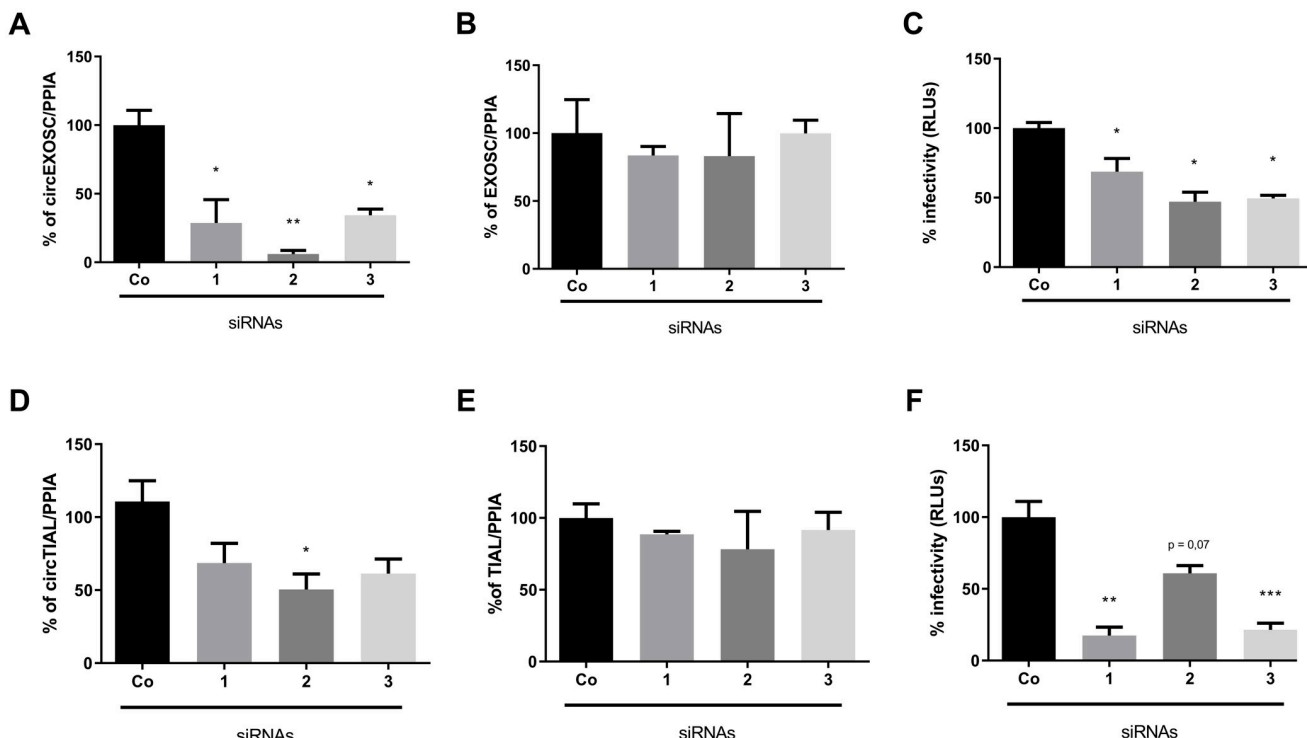

**Fig 2. Effects of circEXOSC and circTIAL depletion on HCV infectivity.** Effects of silencing of circEXOSC1 (**A-C**) with three different siRNAs directed against the back-splice junction on (**A**) circRNA abundances, (**B**), linear mRNA abundances and (**C**) viral infectivity. Effects of silencing of circTIAL (**D-F**) with three different siRNAs directed against the back-splice junction on (**D**) circRNA abundances, (**E**) linear mRNA abundances and (**F**) viral infectivity. All results are related to the expression obtained with the scrambled siRNA (co) and RNA values are normalized to PPIA mRNA abundance. Statistical significance was calculated using a T-test (*represents *p*-value < 0.05). All experiments were performed in triplicates.

statistically significant event (Fig 2D), while displaying no effects on the abundances of their linear counterparts (Fig 2E). However, the diminished effects on HCV infectivity was highly significant (Fig 2F). These findings argue that both circEXOSC and circTIAL are novel pro-viral factors. In contrast, depletion of circRMB39, which is down-regulated after HCV infection (Fig 1C), did not alter HCV RNA abundances (S1A and S1B Fig). However, it cannot be ruled out that lack of phenotype is due to the already downregulation of circRMB39 in infected cells. Similarly, depletion of circPMS1 did not affect HCV RNA abundance (S1C and S1D Fig). Finally, Northern analyses show that siRNA-mediated depletion of randomly selected circRNAs circGALK2, derived from Galactokinase 2 gene, and circENAH, derived from ENAH Actin Regulator gene, did not affect HCV RNA abundances (S1E Fig). These findings argue that changes in HCV infectivity is controlled by the abundance of some, but not of all circRNAs.

### Pleckstrin And Sec7 Domain Containing 3 (PSD3) linear RNA encodes circular circPSD3 RNA that modulates HCV and DENV gene expression

An independent circRNA screen in both the Sarnow and Diez laboratories identified up-regulation of another circRNA, circPSD3 (S1 Table). The altered abundances of circPSD3 in liver cells during viral infection was subsequently and independently verified in both of our laboratories. circPSD3 was chosen for further analysis due to its highly resistance to treatment with ribonuclease RNase R, which selectively degraded single-stranded RNAs, but not circular RNAs. As can be seen in S2 Fig, linear HCV and PPIA, but not circPSD3 or circPTP4A2, is

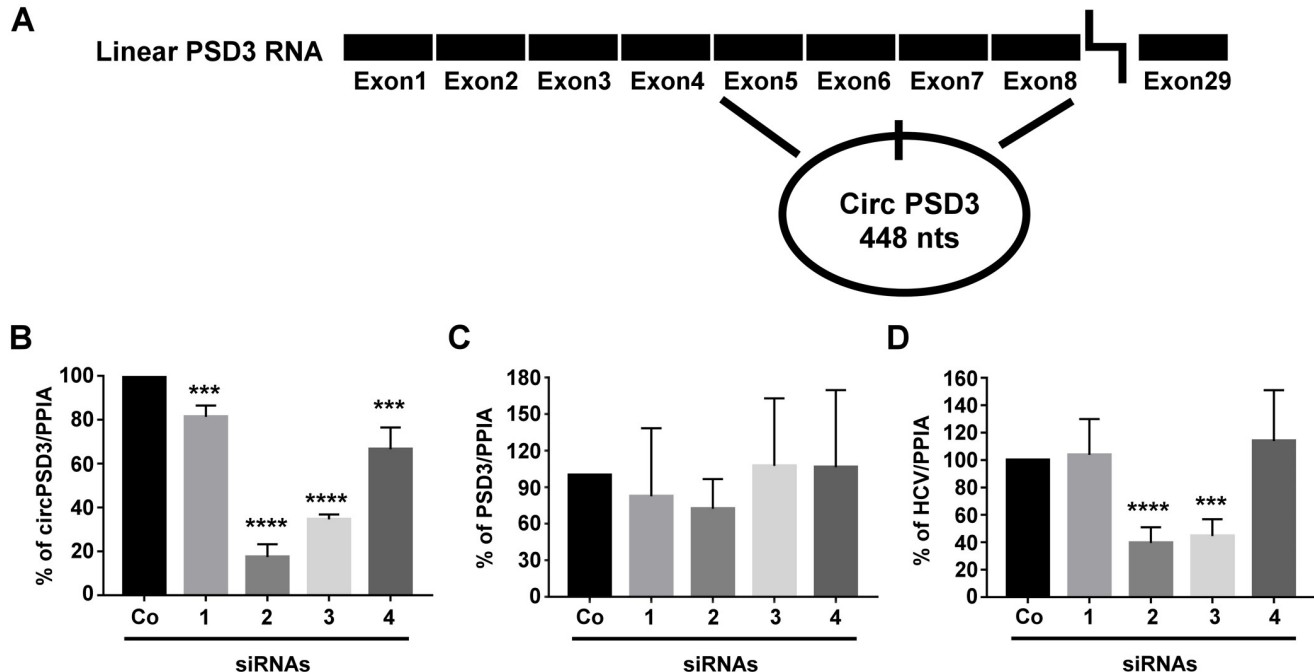

**Fig 3. Effects of circPSD3 depletion on HCV RNA abundance.** (**A**) Diagram of the linear PSD3 RNA and circPSD3 that is generated by a backsplicing event between exons 5 and 8 [43]. (**B-D**) Effects of depletion of circPSD3 RNA by four different siRNAs, which all target the back-spliced junction site in circPSD3 RNA. circPSD3 RNA(**B**), linear PSD3 RNA(**C**) and HCV RNA (**D**) abundances were determined by using RT-qPCR. Results were normalized to PPIA abundances. The data are representative of three independent replicates (***P<0.001 and ****P< 0.0005).

readily degraded by RNAse R. CircPSD3 is even more resilient to nuclease treatment than circPTP4A2 (S2 Fig). CircPSD3 RNA also harbors six predicted binding sites for translation factor eIF4A3 [36]. Importantly, binding of eIF4A3 to PSD3 RNA sequences was verified in high-throughput crosslinking assays [37, 38]. Factor eIF4A3 is of great interest in RNA metabolism due to its association with spliced mRNAs in exon junction complexes that modulate the cellular nonsense-mediated decay (NMD) pathway [39]. Curiously, the NMD pathway is inhibited during HCV infection [40], as well as during infection with mosquito-borne West Nile virus and Dengue virus [41, 42], suggesting that circPSD3 could modulate NMD (more below). Thus, effects of circPSD3 on HCV gene expression were further examined.

Fig 3A displays a diagram of the 29 exon-containing linear PSD3 mRNA and circPSD3 that is derived from a back-splicing event between exons 5 and 8 [43]. Four siRNAs overlapping the exon5/8 junction sequences depleted to various extents circPSD3 RNAs (Fig 3B), with only small effects on the abundances of linear PSD3 mRNAs (Fig 3C). However, there was a strong correlation between loss of circPSD3 and loss of HCV RNA abundance (Fig 3B and 3D). Diminished HCV RNA abundance in circPSD3 siRNA-treated cells could be explained if the four siRNAs affected cell viability and, consequently, HCV RNA expression. However, cell viability assays eliminated this possibility for all four siRNAs directed against circPSD3 (S3 Fig).

To examine whether circPSD3 modulates the infectivity of other RNA viruses, effects on infection rates after circPSD3 depletion were examined in cells infected with DENV, which belongs to the *Flaviviridae*, and Chikungunya virus (CHIKV), which is a member of the *Togaviridae*. For these experiments, we used chimeric DENV and CHIKV viruses expressing luciferase reporter genes. As expected, circPSD3 (Fig 4A and 4D), but not linear PSD3 mRNA (Fig 4B and 4E) was depleted with circPSD3 siRNAs. As was seen in HCV-infected cells, depletion of circPSD3 also diminished DENV infectivity, as quantified by luciferase expression (Fig 4C).

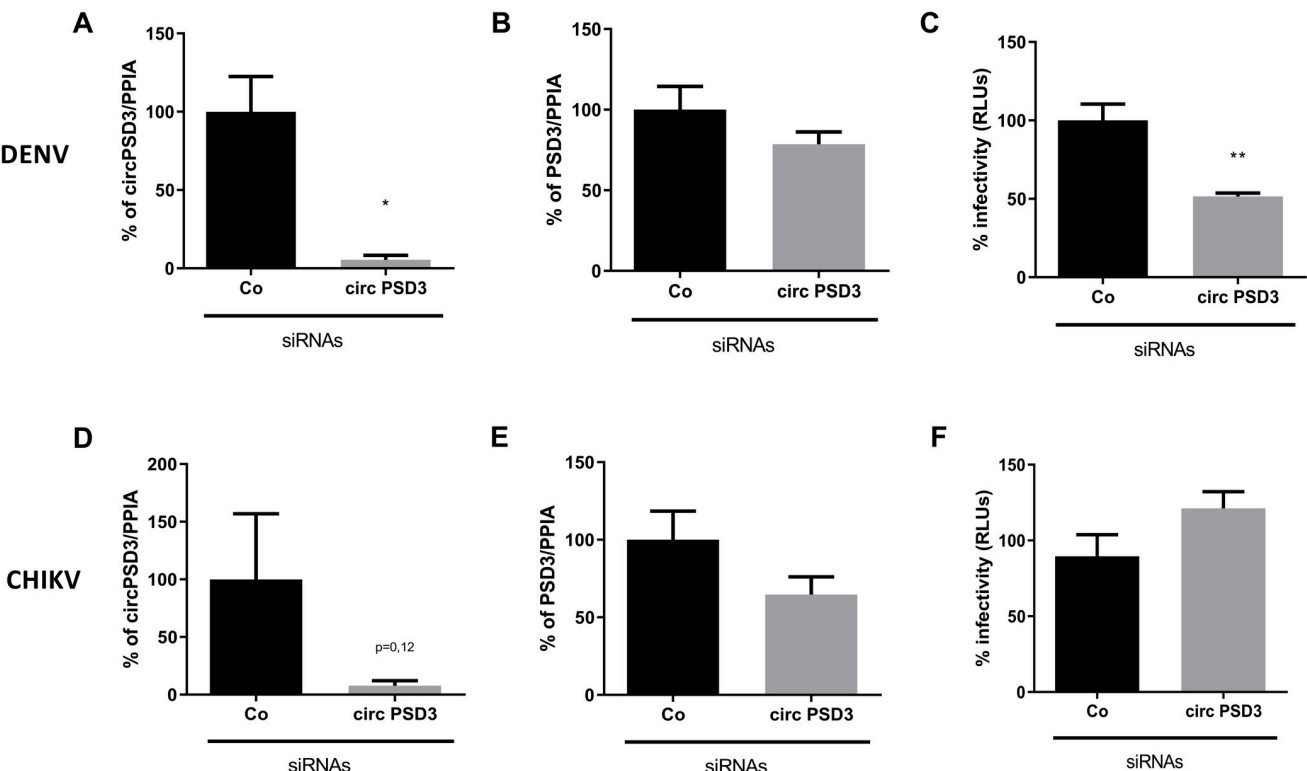

**Fig 4. Effects of circPSD3 depletion on DENV and CHIKV RNA abundances.** (**A-C**) Effects of circPSD3 depletion on luciferase-expressing DENV. (**A**) Circular circPSD3 RNA abundance, (**B**) linear RNA abundance, and (**C**) effects on virus infectivity in circPSD3-depleted cells (circPSD3-siRNA) were compared with those in the control siRNA-transfected cells (Co-siRNA). (**D-F**) Effects of circPSD3 depletion on luciferase-expressing CHIKV. (**D**) Circular circPSD3 RNA abundance, (**E**) linear RNA abundance, and (**F**) effects on virus infectivity were similarly compared between siRNA circPSD3-transfected and siRNA control-transfected cells. Results were plotted as percentage relative to the scrambled siRNA (Co) and are expressed as mean RLU values of three biological replicates (±SEM). Statistical significance was calculated using a T-test (*$p < 0.05$).

However, no effects were seen on CHIKV infectivity (Fig 4F). Furthermore, circPSD3 enhanced extracellular HCV abundance, that was especially significant at a low multiplicity of infection (S4 Fig). These findings argue that circPSD3 has different effects on the growth of distinct RNA viruses, i.e. a pro-viral factor for HCV and DENV, and no pro- or antiviral activity for CHIKV.

## circPSD3 locates to distinct punctate foci in liver cells

To localize circPSD3 in cells, circPSD3 was visualized by fluorescent in situ hybridization, using a back-splicing junction-spanning, bridging oligonucleotide to which fluorophores were attached. Fig 5A shows that circPSD3 located to both the nucleus and cytoplasm, with approximately 3–8 molecules per cell (Fig 5A and 5B). Treatment of cells with siRNAs directed against circPSD3 diminished the number of foci, confirming that detected foci contained circPSD3 molecules. The number of circPSD3 molecules per cell was further quantitated by Droplet Digital PCR. Results showed that approximately ten circPSD3 molecules reside per cell in mock-infected cells and twice as much in infected cells (Fig 5C). The latter finding shows that the puncta were single molecules, but that not all were detected by in situ hybridization. Importantly, these data argue that even low-abundant circRNAs can have significant effects on viral gene expression.

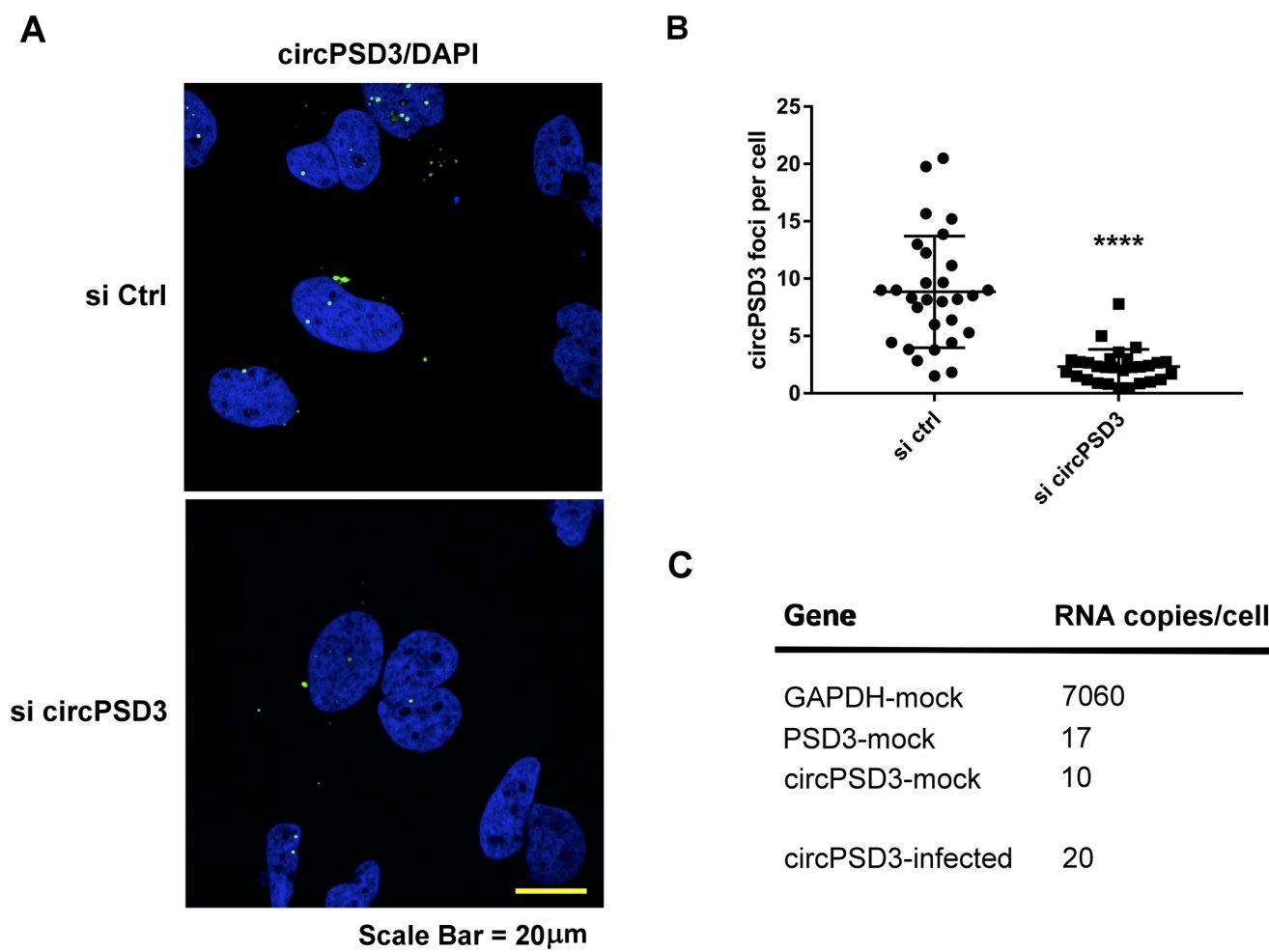

**Fig 5. Localization of circPSD3 RNA in liver cells.** (**A**) Subcellular localization of circPSD3 RNA in Huh7 cells transfected with control siRNA (top panel) or circPSD3 siRNA (bottom panel) at day two after siRNA transfection. Cells were stained with fluorescence-labeled DNA oligonucleotide probes (Alexa Fluor 488), designed to hybridize to the back-spliced junction site in circPSD3 RNA. Nuclei were stained with 4′,6-diamidino-2-phenylindole (DAPI). Confocal microscopic images are shown. The scale bar is 20 μm. (**B**) Quantification of circPSD3 in control-depleted (si ctrl) and circPSD3 RNA-depleted (si circPSD3) cells. Results from three independent transfections are shown. (****p< 0.0001). (**C**) Copy numbers of GAPDH, circPSD3 and PSD3 RNAs in mock- and HCV-infected Huh7 cells. Measurements were obtained by Droplet Digital PCR.

## circPSD3 performs pro-viral functions across genotypes by a post-translational mechanism

To study the mechanism by which circPSD3 enhances HCV RNA abundance, the expression of a J6/JFH1 Renilla luciferase-containing viral RNA was monitored in circPSD3-depleted cells at different times after transfection of viral RNAs. To distinguish effects of circPSD3 depletion on viral mRNA translation and replication, luciferase activities were measured after transfection of chimeric RNAs that contain a mutation in the RNA-dependent RNA polymerase 5B. Fig 6A shows that translation of the viral RNA actually seems to be slightly enhanced in cells treated with circPSD3 siRNAs compared to cells that received control siRNAs. Importantly, these data also show that si-circPSD3 has no off-target effects on the viral RNA genomes. These findings argue that circPSD3 has a minor inhibitory effect on viral mRNA translation and a more pronounced pro-viral effect that occurs either after the steps of entry, i.e. replication or virion assembly.

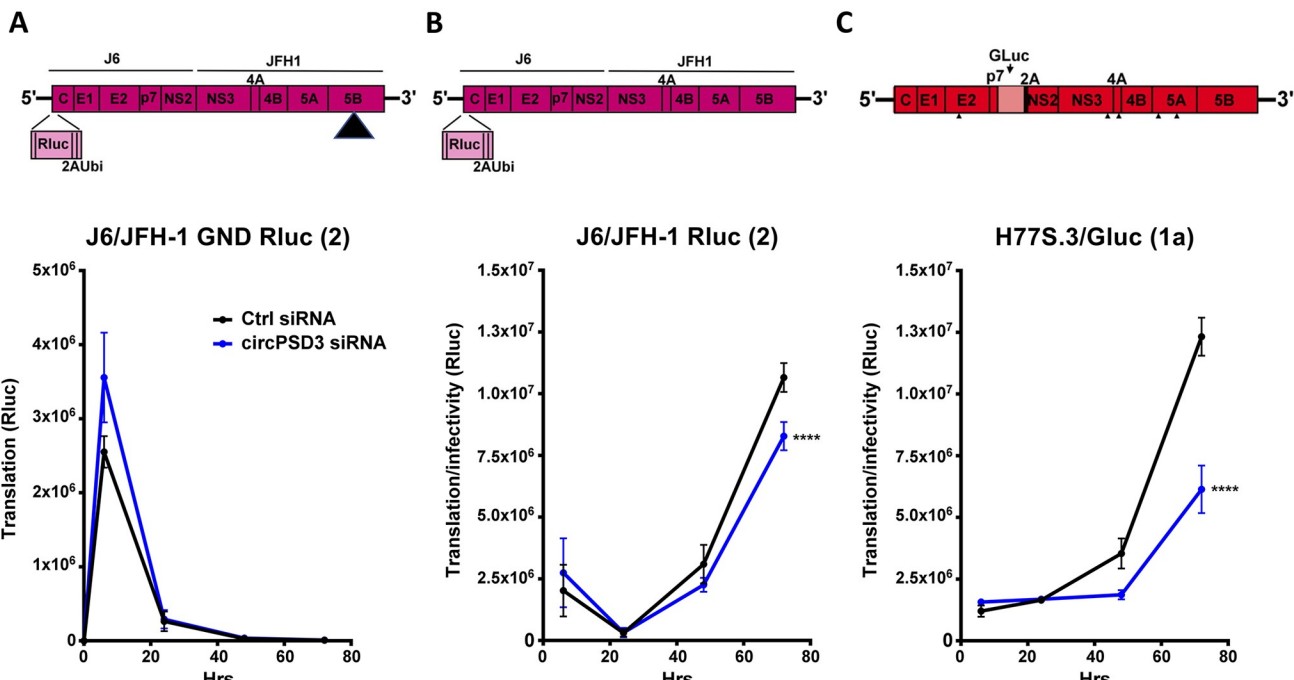

**Fig 6. Effects of circPSD3 RNA depletion on HCV genotype 1a and 2 gene expression.** Structures of the employed infectious viral RNAs, carrying luciferase genes are shown in top panels. **(A)** Effects of circPSD3 RNA depletion on the replication-defective J6/JFH-1 Rluc RNAs. The mutant J6/JFH-1 Rluc replicon has a replication-defective mutation in the NS5B viral polymerase gene (indicated by the triangle). **(B)** Effects of circPSD3 depletion on J6/JFH-1 (type 2) replication was measured by Renilla luciferase activities. **(C)** Effects of circPSD3 depletion on H77S.3/Glu (type 1) replication was measured by Gaussia luciferase production at the indicated time-points. The data are representative of three biological replicates (**p< 0.0081, ****p< 0.0001).

To test whether observed effects of circPSD3 RNA depletion are genotype-specific, RNA abundances in cells transfected with HCV type 2 or 1a genotype were examined. Fig 6B shows a slight, but statistically significant decrease of Renilla luciferase activity in circPSD3 siRNA-treated cells at later times after RNA transfection. Because the JFH1 HCV genome replicates very robustly in liver cells, the effect of circPSD3 depletion on the expression of the more clinically relevant type 1a HCV genotype was examined. Fig 6C shows that there was a significant decrease of luciferase activity at later times in circPSD3 siRNA-treated cells that harbor chimeric Gaussia luciferase-HCV transfected RNA genomes, arguing that circPSD3 has pro-viral functions for at least these two HCV genotypes. Further biochemical analyses are needed to pinpoint the exact step at which viral RNA replication is affected by circPSD3.

## eIF4A3 modulates the cellular nonsense-mediated decay (NMD) pathway independently of circPSD3

As mentioned above, circPSD3 has several verified binding sites for eIF4A3 [36–38], a factor that is essential in the execution of NMD [39]. It is known that HCV inhibits the antiviral function of the NMD in infected cells by a mechanism in which the viral core protein prevents the binding of host protein WIBG (within bgcn homolog) to NMD mediators Y14 and Magoh [40]. Similarly, cells infected with DENV have been shown to inhibit NMD [42] by interactions of the viral core proteins with members of the exon-junction complex that modulates NMD. Thus, we examined the abundances of canonical NMD mRNA substrates SC35D (Splicing Factor SC35) and ASNS (Asparagine Synthetase (Glutamine-Hydrolyzing) [40] in cells infected with a low or high multiplicity of HCV. Abundances of HCV RNA in Fig 7A show

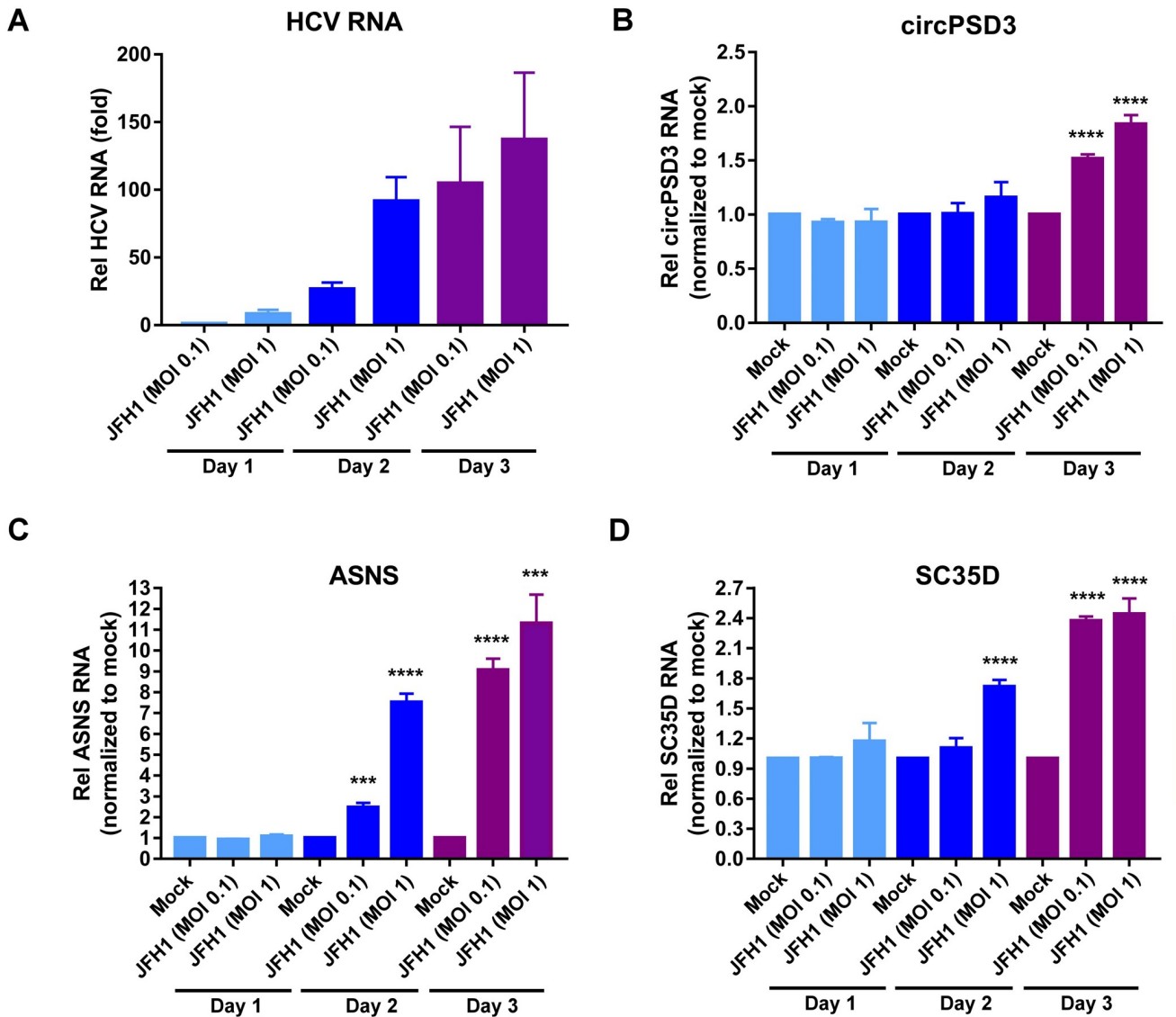

**Fig 7. Kinetics of circPSD3 and NMD mRNA substrate accumulation during HCV infection.** Huh7 cells were infected with JFH-1 virus at MOI of 0.1 or 1. Cells were harvested at day 1, 2 and 3 post-infection. **(A)** HCV RNA, **(B)** circPSD3 RNA, **(C)** ASNS mRNA **(D)** and SC35D mRNA abundances were assessed using RT-qPCR. The RNA abundances were normalized to PPIA mRNA abundance, and then compared to those from mock-infected cells at each day (set to 1.0). The data are representative of three biological replicates (***p<0.001, ****p< 0.0005).

that HCV RNA abundances increased from day one to day three after infection (Fig 7A), with concomitant increase of circPSD3 at day three (Fig 7B). Supporting and extending the observations from Ramage et al. [40] NMD mRNA substrates ASNS (Fig 7C) and SC35D (Fig 7D) increased after three days of infection. While there is a correlation with inhibition of NMD and accumulation of circPSD3 in HCV-infected cells, it is noteworthy that the accumulation of circPSD3 slightly lacks behind the onset of NMD.

It was difficult to examine direct roles circPSD3 and eIF4A3 in NMD, because of low abundance of the circRNA, and the known role for HCV core protein in the inhibition of NMD. Thus, we first examined roles for circPSD3 and eIF4A3 in NMD in uninfected cells using depletion and over-expression approaches. Using specific siRNAs, depletion of circPSD3 led

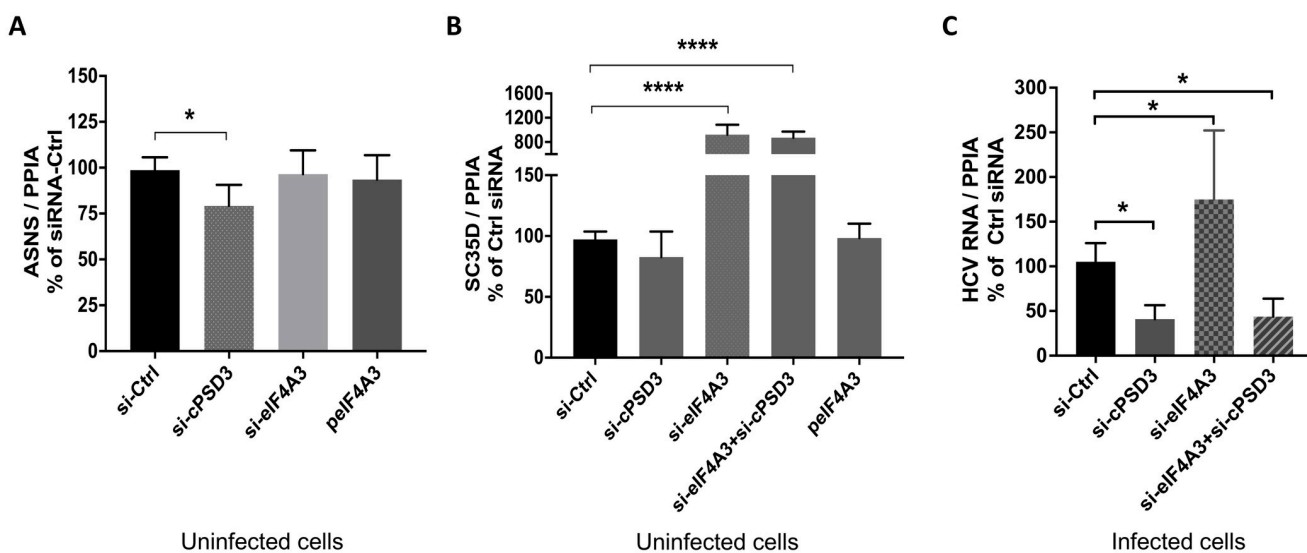

**Fig 8. Effects of circPSD3 RNA and eIF4A3 depletion on NMD substrate abundances and on HCV RNA levels. (A-B)** Huh7 cells were transfected with the indicated siRNAs or plasmid peIF4A3 as indicated. At day three after transfection, RNA abundances of NMD substrates ASNS and SC35D were analyzed using RT-qPCR. **(C)** Huh7 cells were transfected with siRNAs directed against circPSD3 or eIF4A3 or control (Ctrl) siRNAs. At one day post transfection, cells were infected with JFH-1 at 0.5 moi. HCV RNA abundances were determined at day three post infection by RT-qPCR. Results from more than three independent replica are shown (*p<0.05, ***p< 0.001, ****p< 0.0005).

to a slight reduction of NMD substrate ASNS (Fig 8A). This suggested that circPSD3 can modulate the NMD pathway, which may contribute to its pro-viral activity during HCV infection. Interestingly, while ASNS abundance did not change when eIF4A3 was depleted, the abundance of another NMD substrate, SC35D mRNA, was significantly enhanced (Fig 8B). Furthermore, circPSD3 depletion did not affect SC35D abundance, nor was this effect reversed in eIF4A3-depleted cells (Fig 8B). This data suggest that not all NMD substrates need the same the amount of eIF4A3 to trigger their degradation, and that eIF4A3 and circPSD3 likely modulate NMD by different mechanisms. Finally, over-expression of eIF4A3 did not affect abundances of either NMD substrate (Fig 8A and 8B). It is possible that functional heteromeric eIF4A3/Y14/Magoh complexes can not be restored by simply over-expression of eIF4A3, and that circPSD3 and eIF4A3 affect NMD RNA abundances in a more substrate -specific manner.

To examine whether eIF4A3 was involved in the circPSD3-mediated modulation of HCV RNA abundances, eIF4A3 was depleted during HCV infection. Fig 8C shows that HCV RNA abundances increased even though neither eIF4A3 mRNA or protein abundances were altered in infected cells in the presence or absence of circPSD3 (S5 Fig). This result suggests that HCV infection results in a remodeling of the NMD complex [40,42] that is less dependent on eIF4A3, or that eIF4A3 functions as an inhibitor of HCV RNA amplification. Finally, effects of ASNS RNA abundance was examined in infected cells when circPSD3 and eIF4A3 RNAs were depleted. ASNS RNA abundances diminished in circPSD3-depleted, but not in eIF4A3-depleted, infected cells (S6A Fig). In addition, overexpression of eIF4A3 had no effects on HCV RNA abundances (S6B Fig), suggesting, as mentioned above, that functional eIF4A3 complexes were not formed. Control experiments showed that that siRNA depletion and eIF4A3 overexpression approaches were working with high efficiencies (S6C and S6D Fig). These observations from both uninfected and infected cells indicate that circPSD3 and eIF4A3 affect HCV RNA abundances by distinct mechanisms, i.e. eIF4A3 by modulating specific NMD substrate abundances [39] and circPSD3 by enhancing HCV RNA abundances at a post-translational step.

## Discussion

This study examines the global landscape of circular RNAs in HCV-infected cells. It was discovered that specific cellular circRNAs are up- or down-regulated without comparable changes in the abundances of their linear mRNAs. It is doubtful that all identified circRNAs with altered abundances in HCV-infected cells have distinct functions, and some of them might be byproducts of altered transcription and degradation rates caused by the infection. To elaborate, as per our bioinformatics methodology, by definition all differentially produced circRNAs are exhibiting infection-induced abundance changes that cannot be explained by corresponding abundance changes of their linear counterparts. An example can be seen in the circRNA produced by TIAL1 whose abundance increases after infection, whereas its linear transcript is down-regulated. For such cases, an increased transcription rate alone does not suffice to explain the observed behavior, as one would expect both transcript types to show increases in abundance, albeit perhaps not on the same scale. Likewise, a decreased transcription rate should affect both RNAs similarly, as even a more stable circular form should not show any increases in abundance unless the backsplicing frequency has been altered as well. However, if both the transcription rate and the RNA degradation rates are simultaneously increased, the higher stability of circRNA molecules could allow them to benefit from an increased transcription rate and out-live the linear transcripts by avoiding degradation, effectively increasing their abundance while their linear counterpart's abundance would decrease. Another possible explanation for our observations could be that HCV infection indeed alters splicing site preferences and thereby increases the likelihood for backsplicing for all identified genes, although testing this hypothesis would require a detailed analysis of the regulatory mechanisms of backsplicing before and after the viral infection. Similarly, the behavior of circRNAs exhibiting decreased abundance after infection (which we observe much more frequently as can be seen in Fig 1B) might involve alterations of the backsplicing frequency, as the linear transcripts are mostly up-regulated compared to non-infected cells. Thus, unless a circRNA-specific degradation process is induced by HCV infection for these circRNAs, alterations of backsplicing frequencies are a reasonable cause for our observations. Moreover, regardless of the involved mechanism, we are currently unable to directly predict which of the altered circRNAs are in fact functional, as circRNAs can display RNA sequences or structural motifs that are recognized by interacting RNAs or by protein molecules that modulate viral pathogenesis, such as protein kinase PKR (see below).

Several of the identified genes are known to have functions in the life cycles of flaviviruses. For example, TIA1 [44] and TIA1-related/like proteins (TIAR/TIAL-1) [45] act in the nucleus as regulators of transcription and pre-mRNA splicing, and modulate in the cytoplasm the stability and translation of mRNAs, via the formation of stress granules [46]. Curiously, flaviviruses induce the formation of stress granules in infected cells to aid in viral replication, assembly and egress [47–51]. PSD3 has also known functions in hepatitis virus-infected cells and in HCC. PSD3 is overexpressed in HCC, demonstrating tumor-promoting effects of PSD3 [52]. In contrast, liver cells treated with interferon α up-regulate microRNA cluster miR130a/131 that down-regulates PSD3 mRNA and, consequently, PSD3 protein abundances [53]. However, it is unknown whether circPSD3 RNAs have similar functions in normal and transformed cells. Although we have not yet pin-pointed the exact mechanism by which circPSD3 exerts its function on HCV infection, our results support the hypothesis that circPSD3 affects HCV RNA abundance at a post-translational step.

It is doubtful that all identified circRNAs with altered abundances in HCV-infected cells have distinct functions. CircRNAs display RNA sequences or structural motifs that are recognized by interacting RNAs or by protein molecules that modulate viral pathogenesis. Indeed,

the finding by Liu et al. [54] that many circRNAs display small imperfect duplexes that bind to protein kinase R (PKR) and thereby prevent PKR from being activated, suggests that circRNAs may display bulk functions. Thus, enhanced abundances of specific circRNAs in virus-infected cells may have bulk functions in infected cells, or in neighboring cells after vesicle-mediated delivery of circRNAs. *In situ* fluorescent studies would then allow the visualization of cir-cRNAs that moved to uninfected bystander cells under situations where vesicular trafficking is allowed or blocked. Roles for cell-to-cell spread circRNAs are intriguing, because circRNAs are more stable than the linear RNAs from which they are derived from.

With roles for most circRNAs unknown, one could argue that 1–10 copies of a distinct cir-cRNA species has no important roles in gene regulation. However, few copies of circRNAs may cause strong effects on gene regulation, for example, if they have catalytic functions. Fur-thermore, circRNAs have half-lives that are much longer than those of their linear precursor RNAs [55], suggesting that functions of circRNAs may display kinetics that are different from their linear counterparts. Finally, $N^6$-methyladenosine modifications in circRNAs have been shown to impact innate immune responses [56–58]. Thus, differential modification of cir-cRNAs that are up-regulated or down-regulated during viral infections may systemically deliver innate sensors to uninfected bystander cells.

## Materials and methods

### Cell cultures

The human hepatocarcinoma cell lines Huh7, Huh7.5.1 and Huh7 Sec14L2 were maintained in Dulbecco's modified Eagle's medium (DMEM, Invitrogen, Carlsbad, CA), supplemented with 10% heat inactivated fetal bovine serum (FBS) and 10% non-essential amino acids. Kid-ney epithelial monkey cells VERO were maintained in Dulbecco's modified Eagle's medium (DMEM, Invitrogen, Carlsbad, CA) supplemented with 10% heat inactivated fetal bovine serum (FBS) and 10% non-essential amino acids. The baby hamster kidney cells BHK21 were maintained in Glasgow's modified Eagle's medium (GMEM, Invitrogen, Carlsbad, CA) sup-plemented with 10% heat inactivated fetal bovine serum (FBS) and 10% Triptose Phosphate Buffer (TPB). Cells were grown in an incubator with 5% $CO_2$ at 37 ˚C.

### Viral stock productions

Hepatitis C virus D183 (HCV D183) was generously provided by Francis Chisari. To generate HCV D183 stock, Huh-7.5.1 cells were infected with a low MOI (MOI of 0.01). HCV-D183 titer was determined using TCID50. To generate HCV JFH-1 stocks, Huh7 cells were infected with a MOI of 0.01 as previously described [59].

### HCV infection, RNA extraction and RNA-Seq

Six well-plates were inoculated with $2.5 \times 10^5$ Huh7 cells per well. The next day, supernatants were removed and cells were washed twice with phosphate-buffered saline (PBS). Next, cells were infected with HCV-D183 virus at a MOI of 3.4 for 4 hours. The inoculum was removed and replaced with 2 mL of DMEM. 48 hours-post infection, total RNA from infected and non-infected conditions was extracted using TRIzol reagent (Invitrogen), following the manufac-turer's instructions. After treatment with Turbo DNAse (ThermoFisher) of 30 min, the reac-tion was stopped with a second Phenol-Chloroform extraction. 0.5 μg of total RNA was used to perform RNA-Seq. First, ribosomal RNA (rRNA) was depleted using RiboGold (Illumina) and cDNA libraries were generated using the TruSeq Stranded Total RNA kit (Illumina). Sequencing was performed in NextSeq (Illumina) using 2 x 75 cycles of High Output.

## Identification, quantification and validation of circular RNAs

An ensemble of three published methods was used to independently detect circRNAs in every sequenced sample. The detection methods utilized were CIRI2 v2.0.6 based on BWA-MEM (v0.7.16a, parameters: -T 19) [60, 61], find_circ v1.2 based on Bowtie2 (v2.2.9, parameters:—score-min = C,-15,0 –very-sensitive) [62, 63] and CIRCexplorer v1.1.10 based on STAR (v2.5.2b, parameters:—chimSegmentMin 10) [64, 65]. All detection programs were executed with default settings using an hg19 genome build and the RefSeq annotation obtained from UCSC Genome Browser (http://genome.ucsc.edu/). The resulting candidate lists were then merged across all samples using custom Python scripts to create a comprehensive circRNA annotation prior to quantification using SRCP v1.2 [22]. Statistical testing for differential circRNA expression was performed by using R v3.6.0 and the limma-voom package (v3.38.3) and running the voom transformation with quality weights [66]. To increase statistical power for detection of differentially expressed circRNAs, nine available non-infected control samples were used to compare against the HCV-infected triplicates, accounting for possible batch effects during handling. Furthermore, as read counts for circular junctions are sparse in samples without RNAse R treatment and, thus, variability between replicates can be quite high, we substituted the library sizes calculated from the count matrices of circular and linear junctions with library sizes based on full mRNA quantifications performed by kallisto (v0.43.0), which better reflect the actual differences in sequencing depth between samples (lib.size parameter in DGEList) [67, 68]. Thus, after generating an index comprising the GENCODE v24 annotation (hg38 genome build) as well as the HCV H77C/JFH1 genome sequence, we ran kallisto in quantification mode (parameters:—rf-stranded -b 100) and imported the resulting counts via the tximport R package (v1.10.1) [69] prior to limma-voom testing. We then tested for significant differences in the expression of the circular and linear variants of each transcript by utilizing an interaction term (~batch+condition*RNAtype). CircRNAs with FDR < 0.1 were considered as significantly altered compared to their linear isoforms. For candidate selection, we sub-grouped our results based on the fold change directions of both linear and circular isoforms using ggplot2 v3.2.1 for visualization [70].

## Generation of chimeric viruses carrying luciferase reporter genes

Viruses carrying luciferase reporter viruses were used for the silencing studies shown in Fig 2. HCV carrying the firefly luciferase reporter (HCV FLuc) was produced from plasmid pFK-Luc-Jc1 [71]. pFK-Luc-Jc1 was linearized with DNA restriction enzyme MluI (FastDigest) and purified by phenol: chloroform: isoamyl alcohol extraction. Purified DNA was in vitro transcribed using the MEGAscript T7 Transcription Kit (ThermoFisher) according to the manufacturer's protocol. RNA was purified with RNeasy Mini Kit (Qiagen) and electroporated into Huh7.5.1 cells as described previously [72]. DENV-2 strain 16681 carrying a renilla luciferase reporter gene (DENV-RLuc) was generated from pFK-DVs-R2A (kindly provided by Ralf Bartenschlager) [73]. A virus stock was prepared after transfection of in vitro-transcribed RNA into BHK-21 cells. CHIKV carrying gaussian luciferase reporter (CHIKV-Gluc) was produced from an initial stock kindly provided by Dr. Merits (University of Tartu, Estonia). BHK-21 cells were infected with a MOI of 0.1 for 1 hour at 37˚C; after that, the inoculum was replaced with GMEM. The next day, when cytopathic effects were seen, supernatant was harvested. Supernatant was sedimented for 20 min 4˚C at 4000rpm and aliquoted. CHIKV-Gluc titers were measured by plaque assay. All viral stocks were stored at -80˚C.

## Silencing experiments with chimeric, luciferase-carrying viruses

SiRNAs against back-spliced junctions in the circular RNAs were designed using Dharmacon Software: si circEXOSC_1 ATGGAAGGGGGGTTGAAATTTA; si circEXOSC_2 CCTATGGA AGGGGGGTTGAAAT; si circEXOSC_3 GGAAGGGGGGTTGAAATTTATA; si circTIAL_1 GCCGGATATGCTGCCCAAACT; si circTIAL_2 CGGATATGCTGCCCAAACTCC; si circ-TIAL_3 CAAGCCGGATATGCTGCCCAA; si circRBM39-1 AUUCCAGAAGGUAGAAAG AGUdTdT; si circRBM39-2 AGAUUCCAGAAGGUAGAAAGAdTdT; si circRBM39-3 UCCAGAAGGUAGAAAGAGUUAdTdT. Briefly, 4x10$^4$ Huh7 cells per well were seeded in 24 well plates. 24hrs later, the supernatants were removed, cells rinsed twice with PBS and incubated with OptiMEM (ThermoFischer) for 30 minutes. Transfection solutions were prepared using DharmaFECT I (Dharmacon). After 20 min incubation, solutions containing 50nM siRNAs were added to the cells and incubated for 4h at 37˚C, after which the inoculum was replaced with DMEM. The following day, cells were infected with a MOI of 0.05 using viruses carrying luciferase reporters. For HCV-FLuc and DENV-RLuc, cells were incubated with virus for 4h at 37 ˚C, after which the medium was replaced with fresh DMEM. For CHIKV-GLuc, cells were incubated for 1h at 37˚C and then the medium was replaced with fresh DMEM. Two days after infection with HCV and DENV, supernatant was removed, and cells were washed twice with PBS. RNA and lysates were prepared. Cells were harvested after 16 hours of infection with CHIKV-GLuc. As CHIKV-GLuc secretes gaussia luciferase, supernatant was collected and inactivated with UV light. Cells were washed twice with PBS and RNA was extracted as described above. After DNAse treatment, qPCR was performed. To measure firefly luciferase, cells were lysed with 150 μl of Passive Lysis Buffer (PLB1X, Promega) and frozen at -80˚C for at least 10 minutes. The lysate was mixed with 25 μL of Luciferase Assay Reagent II (LAR-II) and after 5 minutes of incubation, luciferase activities were measured in a luminometer (setting time 2 seconds). To measure renilla luciferase (DENV-RLuc), cells were lysed in 150 μl of Renilla lysis buffer (RLB1X, Promega) and frozen at -80˚C. Upon thawing, 4 μl of the lysates were mixed with 20 μl of Renilla Luciferase Assay Buffer and 1/200 of substrate from the Renilla Luciferase assay system (Promega) and measured immediately in a luminometer for 2 seconds. Gaussia luciferase activity in CHIKV-GLuc lysates was measured as in DENV-Rluc lysates (see above).

## Silencing experiments during HCV JFH-1 infection

SiRNAs against back-spliced junctions of cPSD3 (Fig 3) were designed using Dharmacon Software. RNA oligonucleotides were synthesized by Stanford PAN facility (Stanford, CA). The siRNA sequences are as follow: si Control, 5'-GAUCAUACGUGCGA UCAGAdTdT-3'; si circPSD3-1: 5'-ACAUGGCCACCAAUGCUGGGGdTdT-3'; si circPSD3-2: 5'- GGCCACCA AUGCUGGGGGUGAAdTdT-3'; si circPSD3-3: 5'-GAUCUACAUGG CCACCAAUGCdTdT-3'; si circPSD3-4: 5'- CUACAUGGCCACCAAUGCUGGdTdT-3'; sicircPMS: 5'- AUAACAA GC UGCUCUGUUAAAdTdT-3'. For formation of RNA duplexes, 50 μM of sense and antisense strands were mixed in 5 x annealing buffer (150 mM HEPES (pH 7.4), 500 mM potassium acetate, and 10 mM magnesium acetate) to a final concentration of 20 μM, denatured for 1 min at 95˚C, and annealed for 1 h at 37˚C. Huh7 cells (1.25 x 10$^5$) were seeded in 6-well plates and transfected with 50 nM of siRNA duplexes the following day, using Dharmafect I reagent (Dharmacon) according to the manufacturer's instruction. After 24 h post-transfection, the cells were infected with HCV JFH-1 virus at a MOI of 0.1 at 37˚C. After a 5 h incubation, cells were washed with PBS to remove unbound virus, trypsinized, replated in duplicate tissue culture dishes, and harvested at day 3 post-infection.

## Silencing experiments with J6/JFH-1 Rluc genotype 2 and H77S.3/GLuc genotype 1a viruses

Plasmid HCV J6/JFH-1 Rluc genotype 2 contains a renilla firefly reporter, and plasmid HCV H77S.3/GLuc genotype 1a carries a gaussia luciferase reporter [74, 75]. Plasmids were linearized [74] and transcribed using the MEGAscript T7 Transcription Kit (ThermoFisher) according to the manufacturer's protocol. Huh7.5 Sec14L2 cells were transfected with siRNA duplexes at 50 nM final concentration at day 1. The following day, the cells were transfected with 1 μg of in vitro-transcribed J6/JFH-1 Rluc RNA or H77S.3/GLuc RNA using the TransIT mRNA transfection kit (Mirus Bio LLC) according to the manufacturer's protocol. Renilla and gaussia luciferase activities were measured as described [74], using a Glomax 20/20 luminometer for 10 second integration time.

## RNA isolation and quantitative PCR

Cells were washed twice with PBS and total RNA was extracted using TRIzol (Invitrogen) following the manufacturer's protocol. For cDNA synthesis, reverse transcription (RT) was performed using High Capacity RNA to cDNA kit (Thermo Fisher Scientific) following the manufacturer's protocol. RT reactions followed by quantitative PCR was performed using the Power Up SYBR Green master mix, following the manufacturer's protocol (Thermo Fisher Scientific). PCR reaction settings included an initial step of 95 ˚C for 10 min, followed by 40 cycles of 95 ˚C for 15 seconds plus 60˚C for 60 seconds, followed by calculations of relative RNA abundances. Ct values for the gene expression were normalized to the Ct value for an internal control PPIA. The primer sequences for the HCV JFH-1 were: forward 5'- TCTGCG GAACCGGTGAGTA -3' and reverse 5'- TCAGGCAGTACCACAAGGC -3'. The primer sequences were: PPIA: forward 5'- GCGTCTCCTTTGAGCTGTTTGC -3' and reverse 5'- AT GGACTTGCCACCAGTGCC -3'; PSD3: forward 5'- GGCTTCTGAAGGTTGCTGTC -3' and reverse 5'- GCTATGGGCCAGACTCTCAG -3'; circPMS1: forward 5' GAAAGCAGTTTTAC TCAACTGCAA -3' and reverse 5' TAACAGAGCAGCTTGTTATG -3'; SC35D: forward 5'- CGGTGTCCTC TTAAGAAAATGATGTA -3' and reverse 5'- CTGCTACACA ACTGCGC CTTTT -3'; ASNS: forward 5'- GGAAGACAGCCCCGATTTACT -3' and reverse 5' AGCAC GAACTGTTGTAATGTCA -3'; eIF4A3: forward 5' AATCCGCATCTTGGTGAAAC -3' and reverse 5'- CCACTCTTCCCTCTCCACTG -3'.

## Quantitative PCR validations of identified circular RNAs

Total RNA from RNA-Seq experiments and from an independent experiment were reverse-transcribed using SuperScript III (Invitrogen) following the manufacturer's instructions. Next, cDNA was treated with RNAse H (NEB) to minimize RNA contamination. Primers amplifying the predicted back-splice junction sequences in circular RNAs, and predicted contiguous sequences in linear mRNAs were used in PCR reactions containing SYBR Green (ThermoFischer).

## RNA FISH experiments

A customized branched-DNA probe targeting the back-splicing junction of cPSD3 RNA for the ViewRNA Cell Plus assay was obtained from Thermo Fisher. Huh7 cells were transfected with control or circPSD3 siRNA for 2 days. Cells were seeded on the Neuvitro German coverslips with poly-D-lysine coating (GG-12-PDL, Fisher Scientific) in 24-well plates. The RNA signals of cPSD3 were analyzed using ViewRNA Cell Plus Assay kit (Thermo Fisher) according to the manufacturer's protocol. Briefly, cells were fixed and permeabilized for 30 min at room

temperature. The cells were then washed with PBS including RNase inhibitor, hybridized with the target probe in 1:100 dilution in a pre-warmed probe set solution at 40˚C for 2 hours, and then washed with RNA wash buffer solution for six times. PreAmplifier DNA, Amplifier DNA and fluorophore Label Probe were diluted 25-fold. Hybridizations were performed at 40˚C for 1 hour each and then washed with RNA wash buffer for 5 times. The coverslips were then washed with PBS and stained with DAPI in 1:100 dilution in PBS for 5 min at room temperature. After washing with PBS, the coverslips were embedded in Fluoromount-G. Samples were imaged at room temperature (22˚C) with a 20×/N.A.0.60 or a 63×/N.A.1.30 oil Plan-Apochromat objective on a Leica SPE laser scanning confocal microscope (Leica-microsystems). Images were processed with ImageJ (Ver. 1.48, NIH) using only linear adjustments of contrast and color.

### RNase R treatment

JFH-1 infected cells were harvested at day 3 post-infection. Total RNA was isolated with Trizol reagent according to the manufacturer's protocol. The RNA was treated with or without RNase R at 5 U/µg, SuperRNaseIn at 20 U/µl for 2 hrs at 37 ˚C. The reaction was terminated by adding RNA Clean & ConcentratorTM-5 (Zymo Research). It is noteworthy that Super-RNaseIn targets RNases A, B, C, 1 and T1, but not RNase R. Equal amounts of untreated or RNase R-treated RNA (1 µg) were used in reverse transcription reactions using the High Capacity RNA-to-cDNA Kit (Invitrogen), followed by qPCR reactions using SYBR® Green PCR Master Mix (Invitrogen).

### Droplet digital PCR (ddPCR)

For detection of human GAPDH, PrimePCR ddPCR Expression Probe Assay with fluorophore HEX (dHsaCPE5031597) was obtained from Bio-Rad. To detect circPSD3, the following primers were used: forward 5'- TCTACATGGCCACCAATGCT-3', reverse 5'- TCTGGGGTCTCC TTTTCCAA-3'. The probe sequence for circPSD3 was 5'- AGCTTCTAGCCGTGTTGTTTTC ACCCC (with fluorophore FAM/BHQ, Sigma). The primer sequences for linear PSD3 were: forward 5'- ACCATGGAGGAAGGTGGAGA-3', reverse 5'-GAGTGGCAGCCAGTAAGAG G-3'. The probe sequence for circPSD3 was 5'-TGCCATGCCACCCACAAGAGCAGCA (with fluorophore FAM/BHQ, Sigma).

Total RNA was reverse-transcribed using High Capacity RNA to cDNA kit (Thermo Fisher Scientific) following the manufacturer's protocol. The copy number of the cDNA was determined using ddPCR. The ddPCR reaction mixture contained the ddPCRSupermix for Probes (No dUTP) (Bio-Rad), 900 nM forward and reverse primers, 250 nM fluorophore probe and 50 ng cDNA in a total reaction volume of 25 µl. A volume of 20 µl of the PCR reaction was added to the middle row of wells of the droplet generation DG8 cartridge, and a volume of 70 µl of Droplet Generation Oil was added to the left row. After the generation of the droplets (between 14,000–16,000 droplets/well) in a QX200 droplet generator, the droplet mixture (42 µl) was transferred to a PCR plate. The thermal cycling conditions were 95˚C for 10 min, followed by 50 cycles of 94˚C for 30 sec plus 60˚C for 60 sec, with a final 10 min at 98˚C. After PCR amplification, the positive droplets were counted using QX200 droplet reader (Bio-Rad). The data were analyzed using the QuantaSoft Analysis Pro software (Bio-Rad).

### Northern analysis

Total RNA was analyzed by using northern blot as previously described [59].

## MTT viability assay

The cell viability of circPSD3 siRNA treated cells was determined at day 2 post-transfection using Cell Proliferation Kit I (MTT, Roche) as previously described [59].

## Statistical analysis

Statistical analyses were performed with GraphPad Prism 7. A Student's t-test was employed to assess significant differences between the two groups. Error bars represent standard error of the mean.

## Supporting information

**S1 Fig. Effects of HCV-downregulated circRNAs and further control circRNAs on HCV RNA abundances.** Effects of control and three siRNAs directed against circRMB39 (**A,B**) or circPMS1 RNA (**C,D**) on HCV RNA abundance. RNA abundance was determined by RT-qPCR. (**E**) Effects of various circRNA depletions on HCV RNA abundances, examined by Northern blot analyses.
(PDF)

**S2 Fig. Resistance of circPSD3 and circPTP4A2 to RNase R.** Total RNA from JFH1-infected cells was treated with or without RNase R. RNA abundances were analyzed using RT-qPCR. The RNA abundances are compared to RNA abundances from the untreated samples (set to 1.0). circPTP4A2 is derived from protein tyrosine phosphatase 4A2 mRNA.
(PDF)

**S3 Fig. Cell viability of circPSD3 RNA-depleted cells.** The cell viability of control siRNA and four circPSD3 siRNAs were measured at two days after infection. The data are representative of three independent experiments.
(PDF)

**S4 Fig. Effects of circPSD3 depletion on extracellular HCV JFH1 virus production.** Huh7 cells were transfected with non-targeting control siRNAs (siCtrl) or siRNA targeting circPSD3 (si-circPSD3). At one day post transfection, cells were infected with JFH-1 virus at 0.1 moi or 1 moi. Supernatants were collected at three days post infection and viral titers were determined by focus forming assays (FFU).
(PDF)

**S5 Fig. Effects of HCV infection and circPSD3 depletion on eIF4A3 protein and RNA abundances.** (**A**) eIF4A3 protein abundances were measured by Western blot at three days after HCV JFH-1 infection. Three independent experiments are shown. (**B**) eIF4A3 mRNA abundances in siRNA-transfected cells that were further infected with HCV. Mock cells are non-transfected and non-infected cells. Data from RT-qPCR reactions are shown. (**C**) Effects of circPSD3 depletion on eIF4A3 mRNA abundances in uninfected cells. Data from RT-PCR are shown.
(PDF)

**S6 Fig. Effects of eIF4A3 abundances on NMD and HCV infection.** (A) Cells were transfected with siRNA targeting circPSD3 or eIF4A3, or co-transfected with both siRNAs. At one day post transfection, cells were infected with JFH-1 at 0.5 moi. ASNS abundances were measured 3 days post infection by RT-qPCR. (**B**) Cells were transfected with plasmid peIF4A3. At one day post transfection, cells were infected with JFH-1 at 0.5 moi and incubated for 3 days. HCV RNA abundances were measured by RT-PCR. (**C**) Knockdown efficiencies of individual

siRNA transfections on circPSD3 and linear PSD3 RNA abundances. **(D)** eIF4A3 RNA abundances after transfection with siRNA or peIF4A3 plasmid. RNA abundances were evaluated by RT-qPCR after cells were transfected and further infected for 3 days. Data from three independent experiments are shown (* p<0.05; ****p<0.0001).
(PDF)

**S1 Table. List of selected circRNAs.** The table shows the circRNAs used in this study, including gene name, circle name, sizes of circRNAs, linear RNAs, and primers (5'-3') used for the qPCR-based validations.
(PDF)

## Acknowledgments

We are grateful to Dr. Karla Kirkegaard for many helpful comments during the course of this study. We thank Dr. Julia Salzman for critical comments on the manuscript. We thank Fabian Suchy for technical support with the ddPCR experiments and Drs. Alex Lee and Thomas Hansen for bioinformatics support.

## Author Contributions

**Conceptualization:** Sebastian Kadener, Juana Díez, Peter Sarnow.

**Formal analysis:** Tzu-Chun Chen, Marc Tallo-Parra, Qian M. Cao, Sebastian Kadener, Kunlaya Somboonwiwat, Juana Díez, Peter Sarnow.

**Funding acquisition:** Sebastian Kadener, Juana Díez, Peter Sarnow.

**Investigation:** Tzu-Chun Chen, Marc Tallo-Parra, Qian M. Cao, René Böttcher, Gemma Pérez-Vilaró, Pakpoom Boonchuen.

**Project administration:** Sebastian Kadener, Juana Díez, Peter Sarnow.

**Supervision:** Sebastian Kadener, Kunlaya Somboonwiwat, Juana Díez, Peter Sarnow.

**Writing – original draft:** Peter Sarnow.

**Writing – review & editing:** Tzu-Chun Chen, Marc Tallo-Parra, Qian M. Cao, Sebastian Kadener, René Böttcher, Kunlaya Somboonwiwat, Juana Díez, Peter Sarnow.

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
