## [Decision Letter · Decision Letter 0]

19 Feb 2020

Dear Peter,

Thank you very much for submitting your manuscript "Host-derived Circular RNAs Display Proviral Activities in Hepatitis C Virus - Infected Cells" for consideration at PLOS Pathogens. As with all papers reviewed by the journal, your manuscript was reviewed by members of the editorial board and by several independent reviewers. In light of the reviews (below this email), we would like to invite the resubmission of a significantly revised version that takes into account the reviewers' comments. Reviewers clearly recognized the novelty of circular RNA profiling and its role in flaviviral infections but raised concerns regarding the mechanistic aspects of observations. Please respond to these concerns in your revision.

We cannot make any decision about publication until we have seen the revised manuscript and your response to the reviewers' comments. Your revised manuscript is also likely to be sent to reviewers for further evaluation.

Sincerely,

Aleem Siddiqui, Ph.D.

Associate Editor

PLOS Pathogens

Jing-hsiung James Ou

Section Editor

PLOS Pathogens

Kasturi Haldar

Editor-in-Chief

PLOS Pathogens

orcid.org/0000-0001-5065-158X

Michael Malim

Editor-in-Chief

PLOS Pathogens

orcid.org/0000-0002-7699-2064

Reviewer's Responses to Questions

**Part I - Summary**

Reviewer #1: The manuscript submitted by Chen and Tallo-Parra et al represents a very interesting body of work detailing the significance of circular RNAs in viral infection. The experiments are very thoroughly done and well-controlled and the data clearly show a role for circular RNAs in RNA virus infection. The findings are quite novel in that the authors demonstrate a pro-viral role for circRNAs, as well as identify a panel of circRNAs that are responsive to RNA virus infection. They go on to show the pro-viral role of a specific circRNA, circPSD3, which appears to be virus-specific as it does not affect CHKV infectivity. The authors provide a compelling hypothesis that the pro-viral effect of circPSD3 is through the inhibition of nonsense-mediated decay. Previous work suggests that the Flaviviridae family viruses inhibit NMD through multiple mechanisms and this work adds another layer of regulation of this pathway, highlighting its importance in viral infection.

Reviewer #2: The study by Chen et al address the role of host circular RNAs during HCV infection. Our current understanding on the role of circular RNAs during viral infections is very poor. Studies that address this research area are important because it opens a new interface of virus-host interactions and their significance in viral life cycle and pathogenesis. This study suggests that the host derived circular RNA can have a proviral function during the HCV infection. They showed that the circular RNA “circPSD3” has pro-viral function across all HCV genotypes as well as in other Flaviviral infections such as Dengue. In contrast, this circRNA does not play a role in Chikungunya infection. This is a well-designed and methodologically conducted study. However all the conclusions drawn are not adequately supported by the data. They talk about the interaction of circPSD3 with eIF4A3 a factor that is importance for the execution of NMD pathway; however, they did not show the mechanism of how circPSD3 exactly regulates the cellular nonsense-mediated decay (NMD) pathway

Reviewer #3: In this manuscript, the authors profile circRNAs present during HCV infection. circRNAs are made as a product of nuclear back-splicing of pre-mRNAs. While it is clear that they exist, many of their functions are not yet known. They have been profiled during viral infection (EBV, KSHV, and VSV), but very little has been studied related to their mechanisms of action during viral infection. Here, the authors profile these circRNAs in mock and HCV-infected cells, confirm their existence, and then show that a few of them regulate HCV and DENV, but not CHIKV, infection. In particular, they also focus on a circRNA called circPSD3, which seems to have come out of another screen, and show that it also promotes HCV infection. The authors then try to mechanistically link this to a role for this circRNA in blocking NMD, known to inhibit HCV infection. While generally the work is solid, I was a little confused in some aspects, and so my comments below are related to clarifying those aspects, with several experiment suggestions as well.

**Part II – Major Issues: Key Experiments Required for Acceptance**

Reviewer #1: No major issues

Reviewer #2: Major concern of the study that need to be addressed are:

1. How does HCV regulate eIF4A3 levels? Does circPSD3 play a role in negatively regulating eIF4A3. How many copies of circPSD3 are present in HCV infected cells in comparison to mock-infection

2. Does the over expression of eIF4A3 restore the effects of circPSD3 in HCV infections?

3. More validation is required confirm that circPSD3 diminishes NMD in uninfected and, likely, in infected cells, by sequestration of eIF4A3. How circPSD3 effect eIF4A3 at mRNA level or protein is not clear. Immunoprecipitation of eIF4A3 followed by the quantification of circPSD3 in mock and HCV-infected cells can be shown to prove the enhanced interaction between circPSD3 and eIF4A3 in HCV infected cells. Confocal analysis of colocalization between eIF4A3 and circPSD3 can also be performed.

Reviewer #3: 1. One of the novel aspects of this work is the global profiling of the circRNAs during infection. However, Fig. 1 gave very little description of these RNAs. It would be nice to have some more description of them. Are there any defining features that could be described? Do they come from RNAs that are in similar pathways? Are there pre-mRNAs upregulated or downregulated during infection?

2. What explains the changes in abundance of the circRNAs during infection? Is it just that their pre-mRNAs are more upregulated or downregulated?

3. The authors stated in lines that an “independent circRNA screen In HCV-infected cells identified..” It is unclear from the results section if this was an experiment that they did or were the results from published literature? Similarly, directly after they described the results as having been verified in two laboratories, but I’m not sure if they mean in the author’s labs or in published studies?

4. Does circPSD3 affect the production of infectious HCV virions?

5. Does the linear RNA of PSD3 also increase during HCV infection? This could be interesting, especially if it increased prior to the production of the circRNA.

6. The hypothesis set forth by the authors is that the circPSD3 produced during infection is proviral because it inhibits NMD, known to be antiviral. Unfortunately, it appears to be quite difficult to do the direct experiments to test them in context of infection, as the levels of the circRNA are directly related to the amount of virus replication and vice-versa. Therefore, they attempt 2 indirect experiments to address this point. In the first, they show during HCV infection the kinetics of circPSD3 formation and how it relates to the generation of NMD substrates. However, the circRNA appears to be made after NMD has already started, and even when present, NMD still appears to increase. Therefore, it’s not clear from this that the circRNA is slowing down NMD at all. What would happen at Day 4? At that time point, it may be that NMD starts to decrease as a result of circPSD3 upregulation at Day 3?

7. The second set of experiments is based on the fact that others have shown the circPSD3 binds to eiF4A3, a known regulator of NMD, and so they investigate their interplay in the absence of viral infection. Specifically, the authors showed that the depletion of eIF4A3 resulted in increased NMD substrates and so reduced NMD. Based on this idea, I wonder if the authors over-expressed eiF4A3 during HCV infection, a situation which should lead to increased NMD, they could test if the circRNA has less of a regulatory effect on HCV infection? Or what about the reverse - deplete eIF4A3, reduced NMD, circRNA more important for HCV infection? I realize these experiments may be subject to some caveats, as the authors stated, but it seems like it would be important to try to get a little bit closer to proving the concept of the mechanism.

8. Another way to get a little close to the mechanism would be to try to establish the link between circPSD3 and eiF4A3, which was not directly made in this paper. The authors showed that when eIF4A3 is knocked down, more NMD substrates are made (less NMD); in addition they showed that when you deplete circPSD3 fewer NMD substrates are made (more NMD). To show the direct link, what happens if you deplete eIF4A3 -/+ circPSD3? You might expect that if the function of circPSD is mediated by eIF4A3 sequestration to prevent NMD, if you remove eiF4A3, the circRNA should have less of an effect on this process?

**Part III – Minor Issues: Editorial and Data Presentation Modifications**

Reviewer #1: 1. The authors refer to an independent circRNA screen and altered abundance of the circRNA in response to viral infection that was verified by two independent labs. Providing references for these statements or further details would be informative.

2. Although not necessary, it would be useful to larger scientific community if the authors included the the data for all circRNAs altered by HCV infection as supplemental information.

Reviewer #2: (No Response)

Reviewer #3: 1. As you make more NMD substrates, make more circPSD3. But – does it actually inhibit?

2. The logic of why the RNase R experiment was performed for circPSD3 was not clear. Further explanation of why this was done and why it’s important to do would be helpful.

3. In Figure 2, it might be helpful to have the name of the circRNA targeted by the siRNAs located on the bottom panel, as in siRNA: circEXOSC, below the 1, 2, and 3. This seems like it would be especially helpful for the viral infectivity panels, where as it stands now the reader would require the figure legend to known which circRNA was being targeted.

4. For the figure legend for Fig. 5C, it would be helpful if briefly mention how they got from copies/ul to copies/cell. It is unclear why they would change in opposite directions from the copies / ul to the copies / cell depending on the gene (e.g. GAPDH increases, but PSD3 decreases?)

5. In Figure 7, it would be helpful if the authors directly put the MOI on the figure, as opposed to having “low” or “high”. E.g MOI 0.1, MOI 1

6. In Figure 7A, are the CT values for HCV RNA normalized at all or relative to a housekeeping gene? I realize that the CT value of 40 (not detected likely), could make calculating the fold change challenging.

7. Line 341 in methods – “gently” I’m assuming is meant to be generously?

8. Line 400 in methods – “HIKV” – CHIKV?

PLOS authors have the option to publish the peer review history of their article (what does this mean?). If published, this will include your full peer review and any attached files.

Reviewer #1: No

Reviewer #2: No

Reviewer #3: No
---

## [Decision Letter · Decision Letter 1]

28 Jun 2020

Dear Peter

We are pleased to inform you that your manuscript 'Host-derived Circular RNAs Display Proviral Activities in Hepatitis C Virus - Infected Cells' has been provisionally accepted for publication in PLOS Pathogens.

Best regards,

Aleem Siddiqui, Ph.D.

Associate Editor

PLOS Pathogens

Jing-hsiung James Ou

Section Editor

PLOS Pathogens

Kasturi Haldar

Editor-in-Chief

PLOS Pathogens

orcid.org/0000-0001-5065-158X

Michael Malim

Editor-in-Chief

PLOS Pathogens

orcid.org/0000-0002-7699-2064

Reviewer Comments (if any, and for reference):

Reviewer's Responses to Questions

**Part I - Summary**

Reviewer #2: Most of the concerns raised were addressed by the authors in the revised manuscript. The revisions incorporated have significantly enhanced the manuscript quality.

Reviewer #3: The authors have adequately addressed my previous comments, and the new experiments add nicely to the manuscript.

**Part II – Major Issues: Key Experiments Required for Acceptance**

Reviewer #2: (No Response)

Reviewer #3: (No Response)

**Part III – Minor Issues: Editorial and Data Presentation Modifications**

Reviewer #2: (No Response)

Reviewer #3: (No Response)

PLOS authors have the option to publish the peer review history of their article (what does this mean?). If published, this will include your full peer review and any attached files.

Reviewer #2: **Yes: **Gulam Hussain Syed

Reviewer #3: No

---

## [Editor Report · Acceptance letter]

29 Jul 2020

Dear Dr. Sarnow,

We are delighted to inform you that your manuscript, "Host-derived Circular RNAs Display Proviral Activities in Hepatitis C Virus - Infected Cells," has been formally accepted for publication in PLOS Pathogens.

Best regards,

Kasturi Haldar

Editor-in-Chief

PLOS Pathogens

orcid.org/0000-0001-5065-158X

Michael Malim

Editor-in-Chief

PLOS Pathogens

orcid.org/0000-0002-7699-2064